# Validation study of the Indonesian internet addiction test among adolescents

**Kristiana Siste**[1], **Christiany Suwartono**[2], **Martina Wiwie Nasrun**[1], **Saptawati Bardosono**[3], **Rini Sekartini**[4], **Jacub Pandelaki**[5], **Riza Sarasvita**[6], **Belinda Julivia Murtani**[1], **Reza Damayanti**[1], **Tjhin Wiguna**[1]*

1 Faculty of Medicine, Department of Psychiatry, Universitas Indonesia- dr. Cipto Mangkusumo General Hospital, Jakarta, Indonesia, 2 Faculty of Psychology, Atma Jaya Catholic University of Indonesia, Jakarta, Indonesia, 3 Faculty of Medicine, Department of Nutrition, Universitas Indonesia- dr. Cipto Mangunkusumo General Hospital, Jakarta, Indonesia, 4 Faculty of Medicine, Department of Chuld Health, Universitas Indonesia- dr. Cipto Mangunkusumo General Hospital, Jakarta, Indonesia, 5 Faculty of Medicine, Department of Radiology, Universitas Indonesia- dr. Cipto Mangunkusumo General Hospital, Jakarta, Indonesia, 6 Indonesia National Narcotics Board, Jakarta, Indonesia

* twiga00@yahoo.com, tjin.wiguna@ui.ac.id

## Abstract

### Background

Internet addiction is a serious problem that can negatively impact both the physical and mental health of individuals. The Internet Addiction Test (IAT) is the most common used instrument to screen internet addiction worldwide. This study sought to investigate the psychometric properties of an Indonesian version of the IAT.

### Methods

The IAT questionnaire was made the focus of forward translation, expert panel discussions, back translation, an item analysis (30 subjects), a pilot study (385 subjects), and field testing (643 subjects). Factor structure was analysed by exploratory (EFA) and confirmatory factor (CFA) analyses, whereas reliability was measured with Cronbach's alpha coefficient.

### Results

Factor analysis revealed that the Indonesian version of IAT, consisted of 3 domains, and had good validity ($\chi 2$ p < 0.001; RMSEA = 0.076; CFI = 0.95; SRMR = 0.057, and AIC = 784.63). The Cronbach's alpha score is 0.855. A significant association was also observed between the level of internet addiction with gender (p = 0.027) and the duration of internet use per day (p = 0.001).

### Conclusion

The Indonesian version of IAT provides good validity and reliability in a three-dimensional model. Therefore, it can be utilised as a tool for screening internet addiction in the Indonesian population.

**Data Availability Statement:** The data that support the findings of this study are available from the Faculty of Medicine, Universitas Indonesia, but restrictions apply to the availability of these data,

which were used under license for the current study, and so are not publicly available. Data are however available from the authors upon reasonable request and with permission of Faculty of Medicine, Universitas Indonesia. Data can be accessed through Dr. Khamelia, SpKJ (K), a Research Coordinator at the Department of Psychiatry, Faculty of Medicine Universitas Indonesia-dr. Cipto Mangunkusmo General Hospital, Jakarta-Indonesia (email: khameliapsi@gmail.com).

**Funding:** This study received financial support from University Indonesia Research Grant for TADOK (Grant for student's research). The funders had no role in the design, data collection, analysis, interpretation or write-up in this study.

**Competing interests:** The authors have declared that no competing interests exist.

**Abbreviations:** AAP, American Academy of Pediatrics; AGFI, Adjusted Goodness of Fit Index; AIC, Akaike Information Criterion; CFA, Confirmatory Factor Analysis; CFI, Comparative Fit Index; df, Degree of Freedom; DSM-IV, The Diagnostic and Statistical Manual of Mental Disorders, 4th editio; EFA, Exploratory Factor Analysis; FGD, Focus Group Discussion; JHS, Junior High School; IA, Internet Addiction; IAT, Internet Addiction Test; NNFI, Non-Normed Fit Index; RMSEA, Root Mean Square Error of Approximation; SD, Standard Deviation; SEM, Standard Error of Measurement; SHS, Senior High School; SRMR, Standardised Root Mean Square Residual; SPSS, Statistical Package for the Social Sciences; WHO, World Health Organization; $X^2$, Chi-square; $X^2$, Chi-square divided by degrees of freedom.

## Background

The internet has become a necessity in everyday life and is immensely utilised in almost all aspects of people's lives. There was a dramatic increase in the proportion of individuals using the internet from 0.9% in 2000 to 17.1% in 2014 [1]. Based on data from the Indonesia Internet Service Provider Association, the number of internet users in Indonesia has reached 143 million people, becoming the highest number of internet users in the South East Asia region [2].

Despite the benefits that the internet offers such as easy access to unlimited information, limitless communication, and entertainment, its excessive use can lead to addiction [3,4]. Internet addiction (IA) is defined as a pattern of excessive use of internet networks accompanied by poor self-control and constant obsessive thoughts of maladaptive internet use. [5] The term 'internet addiction' was agreed upon for use by psychiatrists given the similarities between its symptoms and symptoms of addiction caused by substances [6]. Internet addiction has been categorized into generalized internet addiction (GIA) and specific internet addition (SIA) due to their different mechanism [7–10]. Internet addiction in this study referred to GIA, which is defined as a general and multidimensional behavioral pattern of internet overuse that causes significant consequences in an individual's life [10]. A previous study indicated that 6% of the world's population or approximately 182 million people experience internet addiction [11]. In addition, prevalence of internet addiction among child and adolescents was high. One study noted that 13.5% child and adolescents experienced internet addiction [12]. Internet addiction is a troubling condition as it can lead to physical and mental health impairments (e.g. cognitive impairment, obesity, and sleep problems) [13–18]. Thus, prompt diagnosis and immediate treatment should be effectively ensured.

Several instruments have been used to identify internet addiction, e.g. Internet Addiction Test (IAT) [5,19]; Persian Internet Gaming Disorder Scale-Short Form (IGDS-SF) [20], Bergen Social Media Addiction Scale (BSMAS) [21], Internet Disorder Scale (IDS-15) [22]. The most common and widely used instrument being the Internet Addiction Test (IAT). The IAT was created by Kimberly Young in 1998 as an instrument to diagnose internet addiction. It was developed from the pathological diagnosis criteria for gambling listed in *the Diagnostic and Statistical Manual of Mental Disorders*, 4th edition (DSM-IV). This questionnaire consists of 20 questions in English regarding problematic behaviors that occur due to excessive internet use. It adopts a Likert scale from 0–5 with Cronbach's alpha value 0.83–0.91 [5,19]. Scores obtained from the IAT are grouped into four categories: normal (0–30), mild IA (31–49), moderate IA (50–79), and severe IA (80–100). The IAT has been widely translated and validated by various countries and has proved to consist of good internal validation values [23–31].

The IAT has also been used in Indonesia; unfortunately, however, the questionnaire has only been translated into the Indonesian language and has not been examined for its psychometric properties. Thus, this study seeks to bridge this gap by assessing the reliability and validity of the Indonesian version of the IAT by analysing its factor structure. The objective of this study is to translate Young's IAT into the Indonesian language and validate the Indonesian version of IAT.

## Methods

### Participants

This study was conducted at nine randomly selected schools from 39 secondary schools in Jakarta that extensively cooperated with the Department of Psychiatry, Cipto Mangunkusumo Hospital, Faculty of Medicine University of Indonesia. The schools consisted of junior high schools (JHS) and senior high schools (SHS) and were also varied in terms of being public,

private, vocational, and religious schools. Cluster random sampling was used to select the representative of each group of schools to participate in this research. Various types of schools were used in this study to represent the diversity of all possible student types.

This study involved students aged 12-18 years old from several JHS and HS in Jakarta. The number of samples available for the IAT item analysis was 30 (15 JHS students and 15 SHS students). The pilot study of the IAT, however, used 385 subjects (145 JHS students and 240 SHS students) and 643 subjects (333 JHS students and 310 SHS students) for the psychometric evaluation study. All participants were selected through stratified random sampling. All participants and their parents or legal guardians in this study were informed of the study protocol verbally and signed an informed consent form.

This study was approved by the Research Ethics Committee of the Faculty of Medicine of Universitas Indonesia - Cipto Mangkusumo Hospital, Jakarta, Indonesia.

## Instruments

The instrument used in this study is the Internet Addiction Test (IAT) developed by Kimberly Young to assess the problems resulting from excessive internet use. The IAT is a self-report instrument using a five-point Likert scale. It is a unidimensional (one-factor structure) questionnaire consist of 20 items that measures psychological dependence, compulsive use, and withdrawal symptoms. The total scores of IAT are subsequently categorised into four groups to determine the severity of internet addiction: normal (0 – 30), mild internet addiction (31 – 49), moderate internet addiction (50 – 79), and severe internet addiction (80 – 100) [5,19].

## Procedures

The IAT was adopted by considering transcultural aspects. The process of this study was in accordance with the process from the guidelines of the World Health Organization (WHO) [32]. The adaptation steps commenced with forward translation, in which the instrument was translated from English into Indonesian by two independent translators whose mother tongue is Indonesian. The Indonesian version of the instrument was subsequently assessed by three experts, including an addiction psychiatrist, a child and adolescent psychiatrist, and an addiction psychologist in order to determine whether the translation results' content it suitable for being adapted into local conditions. The result was then translated into English (back translation) by an independent translator whose mother tongue is English. It was ensured that the translator was not exposed to the original questionnaire beforehand. Following this, the result of the back translation was shared with the original questionnaire creator, Dr. Kimberly Young from Net Addiction, the Center of Internet Addiction, for reviewing the contents of the questionnaire. Then, one round of item analysis was conducted on 15 JHS students and 15 SHS from seven selected schools through the focus group discussion method to determine the comprehensibility and efficiency of the instructions and terms used in the questionnaire. Experts' judgment was requested later. Next, a pilot study was conducted from the instrument produced in the previous stage; in this, the instrument was distributed among 145 JHS students and 240 SHS students from eight selected schools. At this stage, the internal consistency value (the value of Cronbach's alpha) was obtained from the IAT. Following this, psychometric properties' evaluation was conducted with 643 students from nine selected schools in a field test. Exploratory Factor Analysis (EFA) and Confirmatory Factor Analysis (CFA) was also conducted to examine the factor structure and the appropriateness of the factors respectively. Additionally, we also posed questions regarding the initial stages of internet use, the duration of time spent using the internet every day, and the aims of using the internet. Thus, the

relationship between the level of internet addiction that was determined by IAT scores and these factors can be determined as well.

## Statistical analysis

The validity of the IAT's contents were calculated using the internal consistency value (Cronbach's alpha), inter item correlation by Pearson correlation, and EFA by utilizing the IBM Statistical Package for the Social Sciences (SPSS) version 22 for Windows software. Extraction method used for EFA analysis was based on eigenvalue (eigenvalue $\geq$ 1) and by observing the scree plot. The rotation method used was orthogonal rotation/varimax. Meanwhile, CFA was assessed using Linear Structure Relations (Lisrel) version 8.8. In the study, CFA was conducted to confirm the suitability of the IAT's factor structures obtained in the previous EFA. The suitability of the models was based on several parameters such as the p-value of the chi-square test > 0.05, Root Mean Square Error of Approximation (RMSEA) < 0.06, Comparative Fit Index (CFI) $\geq$ 0.9, Standardised Root Mean Square Residual (SRMR) < 0.08, Non-Normed Fit Index (NNFI)/Tucker Lewis Index (TLI) > 0.95, Adjusted Goodness-of-Fit Index (AGFI) > 0.95, and had lower Akaike Information Criterion (AIC) [33].

Detailed statistical analysis for each step of this study are as follows:

1. Item analysis (30 subjects) was analyzed by calculated inter item correlation (Pearson correlation), reliability testing (internal consistency; Cronbach's alpha), and focus group discussion.

2. Pilot study (385 subjects) was analyzed by calculated inter item correlation (Pearson correlation), reliability testing (internal consistency; Cronbach's alpha), and Exploratory factor analysis (EFA)

3. Field testing (643 subjects) was analyzed by Confirmatory factor analysis (CFA)

Recalibration of cut-off was performed if they were any items which were dropped. We used the same method adopted by Karim et al. in Bangladesh (2014) [33] as a reference to recalibrate new cut-off if there were any modifications to the total number of items.

Chi-square analysis was also carried out using SPSS to identify the association between the addiction level and several factors such as age, gender, age of first internet use, and duration and aim of internet use. A significant association was determined as p-values $\leq$ 0.05.

## Results

### Forward translation, expert panel discussion, and back translation

After the IAT questionnaire was translated into Indonesian, it was reviewed by three experts in a panel discussion. The terms "online" and "offline" that had been preserved in the English language by the translators were replaced with their parallel Indonesian counterparts. The term "depression," however, was retained since it is a considerably well-known term among teenagers. The revised questionnaire from the expert panel discussion was translated back into English. The results of the backward translation were then shared with the original questionnaire creator, Dr. Kimberly Young from Net Addiction, the Center of Internet Addiction and approval was obtained.

### Item analysis

An item analysis was conducted to gather the construct validity and reliability of the IAT questionnaire through a focus group discussion (FGD). A total of 30 students (15 JHS students and

**Table 1. Characteristics of the research subjects.**

| Variable | Item Analysis Frequency (%) | Pilot Study Frequency (%) | Psychometric Properties Evaluation Frequency (%) |
|---|---|---|---|
| Gender | | | |
| **Male** | 9 (30) | 183 (47.5) | 298 (46.3) |
| **Female** | 21 (70) | 202 (52.5) | 345 (53.7) |
| Age | | | |
| **Early adolescent (11–16 years old)** | 27 (90) | 329 (85.5) | 561 (87.2) |
| **Late adolescent (17–25 years old)** | 3 (10) | 56 (14.5) | 82 (12.8) |
| Education | | | |
| **Junior high school** | 15 (50) | 145 (37.7) | 318 (49.5) |
| **Senior high school** | 15 (50) | 240 (62.3) | 325 (50.5) |
| Age of first internet use | | | |
| **$\leq$ 8 years old** | N/A | 362 (94) | 132 (20.5) |
| **> 8 years old** | N/A | 23 (6) | 511 (79.5) |
| Duration of internet use | | | |
| **$\leq$ 20 hours/week** | 9 (30) | 83 (21.6) | 212 (32.9) |
| **> 20 hours/week** | 21 (70) | 302 (78.4) | 431 (66.9) |
| Aim of using internet | | | |
| **Education** | 7 (23.3) | 148 (23.0) | 134 (20.8) |
| **Entertainment** | 8 (26.7) | 124 (19.3) | 56 (8.7) |
| **Game online** | 7 (23.3) | 192 (29.8) | 109 (17.0) |
| **Social media** | 8 (26.7) | 176 (27.4) | 343 (53.3) |
| **Communication** | 0 (0) | 3 (0.5) | 1 (0.2) |
| Internet addiction | | | |
| **Normal (IAT scores < 45)** | 25 (83.3) | 323 (83.9) | 545 (84.8) |
| **Addiction (IAT scores $\geq$ 45)** | 5 (16.7) | 62 (16.1) | 98 (15.2) |

15 SHS) participated in the study. Nine females and 21 males took part in the study with ages ranging from 12–18 years. The characteristics of the subjects are listed in Table 1.

All participants were asked to fill in the Indonesian version of the IAT that consisted of 20 items before the FGD. During the FGD, the students made some suggestions pertaining to the terms used in the questionnaire to make them more familiar to teenagers. Changes were made in several statements without altering their intended meaning. The students also did not know about the term "log in" because currently, electronic devices do not require to be logged into. The term "log in" was therefore changed to "online." The term "online" was also considered for replacement with the term "playing internet" since being online does not necessarily indicate active internet use. The word "couple" is also not suitable for teenagers; therefore, it was replaced with "family," "friends," or "closest person." The term *"pasangan"* (lit. couple) is also considered to imply a romantic relationship and therefore was replaced with *"orang - orang terdekat"* (lit. relatives). "Work productivity" is also not applicable to teenagers and was replaced with "academic achievement."

The Pearson correlation test was carried out between each item with the total score to assess the validity of the IAT questionnaire. It was observed that all items were valid since the correlation between items was above 0.3 (ranging from 0.419 to 0.788). The questionnaire also exhibited very good reliability with an α-Cronbach value of 0.913.

Following the item analysis, the results were discussed by the three experts. The altered terms are listed in Table 2. Next, a consultation was held with Indonesian language experts from the Faculty of Literature, University of Indonesia. Minor modifications were made in

**Table 2. Results of exploratory factor analysis.**

| Items | | Factor Loading | | |
|---|---|---|---|---|
| | | Salience | Neglect of duty | Loss of control |
| 3 | How often do you choose internet enjoyment over intimacy with your family, friends, or the person closest to you? | 0.644 | | |
| 11 | How often do you find yourself planning when you will play on the internet again? | 0.569 | | |
| 12 | How often do you fear that life without the internet would be boring, empty, and joyless? | 0.688 | | |
| 13 | How often do you get angry, yell, or act annoyed when someone disturbs you while you are playing on the internet? | 0.578 | | |
| 15 | How often do you continuously think about the internet while you are not playing on the internet or fantasize about playing on the internet? | 0.586 | | |
| 19 | How often do you choose to use more time to play on the internet over going out with other people? | 0.539 | | |
| 20 | How often do you feel depressed, unstable, or nervous when you are not playing on the internet and that disappears once you are back to playing on the internet? | 0.617 | | |
| 2 | How often do you neglect household chores to spend more time playing on the internet? | | 0.499 | |
| 6 | How often do your grades or school-work suffer due to the amount of time that you spend to play on the internet? | | 0.845 | |
| 8 | How often does your school performance or assignment suffer due to the internet? | | 0.850 | |
| 14 | How often do you not sleep due to playing on the internet all night long? | | 0.489 | |
| 17 | How often do you try to reduce the time you spend playing on the internet and then fail? | | 0.413 | |
| 1 | How often do you find that you play on the internet for longer than intended? | | | 0.449 |
| 4 | How often do you form new friendships with fellow people who play on the internet? | | | 0.577 |
| 9 | How often do you close yourself off or behave in a secretive manner when someone asks you what you do when playing on the internet? | | | 0.676 |
| 10 | How often do you cover disturbing thoughts with pleasant thoughts about the internet? | | | 0.637 |
| 16 | How often do you say "just a minute" when playing on the internet? | | | 0.474 |
| 18 | How often do you try to hide the amount of time you really spend playing on the internet? | | | 0.552 |
| **Eigenvalues** | | 5.412 | 1.708 | 1.231 |
| **Variance percentage** | | 30.069 | 9.487 | 6.836 |
| **Reliability** | | 0.761 | 0.691 | 0.686 |

some sentences in accordance with adolescents' understanding level and in order to better emphasise the idea.

## Pilot study

In the pilot study, the assessment of IAT validity and reliability was conducted on 385 subjects (145 JHS and 240 HS students). The majority of subjects recruited for the pilot study were female (52,5%) and early adolescents (85,5%). The subjects who participated were different from those involved in the focus group discussion. The ratio of male and female participants was 47.5%:52.5%. The result revealed that most participants used the internet anywhere and by using a modem (54.8%). The majority surfed the internet for 4–8 hours per day (61%); however, 5.7% of the participants used the internet more than 8 hours per day. The aim of using the internet was mostly to play online games (81.8%). From 20 items in the IAT questionnaire, the corrected item-total correction test was conducted. It was revealed that the values for item-total correlations ranged from (0.206–0.577). Item number 7—"How often do you check your email before doing the other activities that you need to do?"—was found to have a correlation value of 0.206, thereby indicating poor reliability. Hence, the item was deleted.

An EFA was next performed in the 19-item IAT questionnaire. From the first EFA, item number 5 "How often do people in your life complain about the amount of time that you

spend to play on the Internet?"—was found to have loading factors < 0.4, thereby indicating poor validity. This item was hence deleted.

Next, the second EFA for the 18-item IAT was conducted. It showed four factors or domains with eigenvalues more than one and explained 52.557% of the total variance. The grouping of the factors was based on the highest loading factor within the particular domain with a minimal value of the loading factor equal to or more than 0.4. The results showed that each item has a satisfactory loading factor (> 0.4). However, domain 3 consisted of only two items and did not fulfill the minimum requirement of three items. Consequently, we performed a re-run of the analysis and specifically asked for three components.

Subsequently, the third EFA was run. Unlike the first and second EFA, eigenvalue was not used to determine the domain in the third EFA since the domains had been decided from the beginning by the determining extract factor. The third EFA revealed three domains and all items had a loading factor > 0.4. The factor loads related to the 18 items ranged from 0.449 to 0.850, thereby indicating that these questions were sufficiently qualified to be included in the test. The three domains, along with the factor loadings, are listed in Table 2. The three domains were titled salience, neglect of duty, and loss of control.

The internal correlation was retested, and the result showed a good correlation (above 0.3) for each of the 18 questionnaire items. Values for the item-total correlations ranged from 0.316 to 0.576. Moreover, the internal reliability coefficient was 0.862.

## Field testing

Following the pilot study, a total of 643 subjects (333 JHS students and 310 SHS students) participated in the field test for the psychometric evaluation study. Most subjects recruited for field testing was female (53.7%) and in early adolescent phase (87.2%). The characteristics of the participants are described in Table 1. To note, the first 385 respondents in the dataset were overlapping and employed within EFA (N = 385) and CFA (N = 643).

Two models were assessed in this study: the first model used the original version of IAT (one domain, 20 items) while the second model is in accordance with the EFA results (three domains, 18 items).

CFA's first model resulted $\chi2$ (df = 152, p < 0.001) = 488.05 and $\chi2$/df = 3.21 with RMSEA = 0.059, CFI = 0.97, SRMR = 0.046, and AIC = 604.05. While the second model generated $\chi2$ (df = 126, p < 0.001) = 479.50 and $\chi2$/df = 3.81 with RMSEA = 0.066, CFI = 0.96, SRMR = 0.048, and AIC = 596.50 (Table 3). The results of each model were subsequently compared.

The first parameter that evaluated was the Akaike Information Criterion (AIC). The second model was.more efficient due to its lower AIC. Following this, other parameters were also compared and it was discovered that the second model exhibited a higher value than the first model in all goodness of fit indices. Hence, the second model was the preferred model in this study. The results of both CFA are given in Figs 1 and 2.

**Table 3. Reliability coefficient of each domain.**

| Domain | No. of Items | Alpha | SEM | Mean | SD | Corrected Item Total Correlations |
|---|---|---|---|---|---|---|
| Salience | 7 | 0.761 | 2.150 | 11.465 | 4.399 | 0.418 – 0.570 |
| Neglect of duty | 5 | 0.691 | 1.944 | 9.073 | 3.498 | 0.369 – 0.541 |
| Loss of control | 6 | 0.686 | 7.054 | 12.589 | 4.237 | 0.313 – 0.480 |
| Total | 18 | 0.855 | 3.870 | 33.127 | 10.165 | 0.317 – 0.574 |

SEM = Standard Error of Measurement.

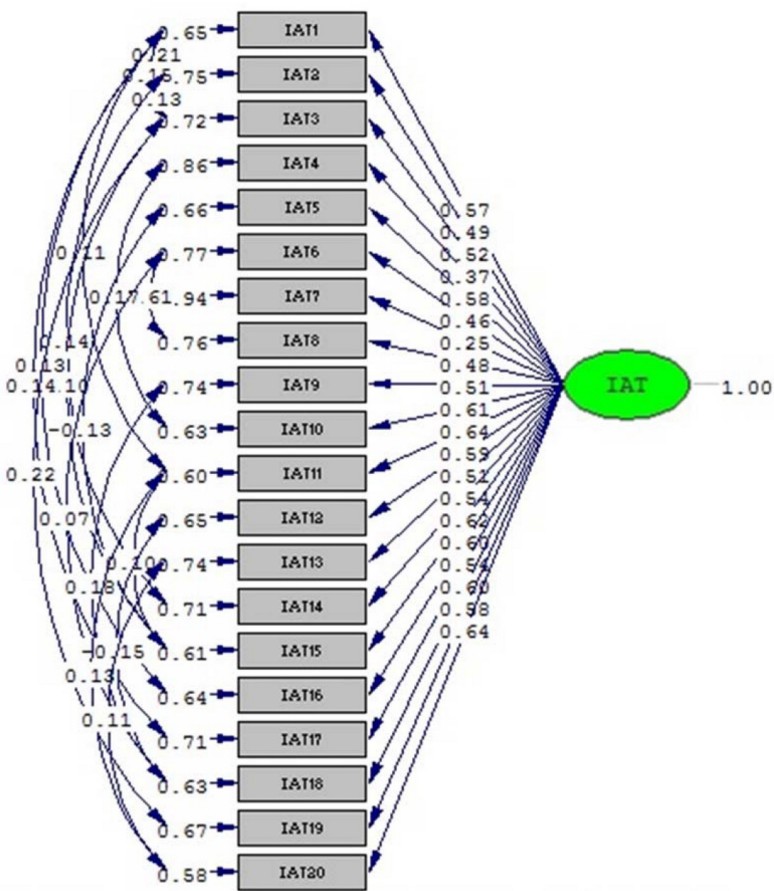

Chi-Square=488.05, df=152, P-value=0.00000, RMSEA=0.059

**Fig 1. Confirmatory factor analysis result of first model IAT.**

Cronbach's alpha coefficient was also calculated to measure the reliability of the instrument. The reliability of model 2, which consisted of three domains and 18 items, was analyzed. Values for item-total correlations ranged from 0.317 to 0.574 with the internal consistency value of the Cronbach alpha coefficient being 0.855. The values of the reliability coefficient for each domain are listed in Table 4.

As two items were dropped, we recalibrated the cut-off scores for IAT. The level of internet addiction was determined through a new cut-off of IAT scores (Table 5) [7].

This study also analysed the relationship between the extent of internet addiction, age, gender, age of first internet use, duration, and aim of internet use. A significant association was revealed between the extent of internet addiction and gender $(x2(df) = 4.921(1), p = 0.027, OR = 1.669, CI = 1.081–2.577)$ and duration of internet use per week $(x2(df) = 5.094(1), p = 0.024, OR = 0.545, CI = 0.329–0.905)$. Meanwhile, no significant association was observed between the extent of internet addiction and age, aim, and age of first internet use $(p > 0.05)$.

The level of internet addiction was determined through IAT scores (normal, mild, moderate, and severe addiction) [7]. The cut-off scores for categorising internet addiction were formulated in the 18-item Indonesian version of the IAT since it exhibited better psychometric properties than the 20-item version (Table 5).

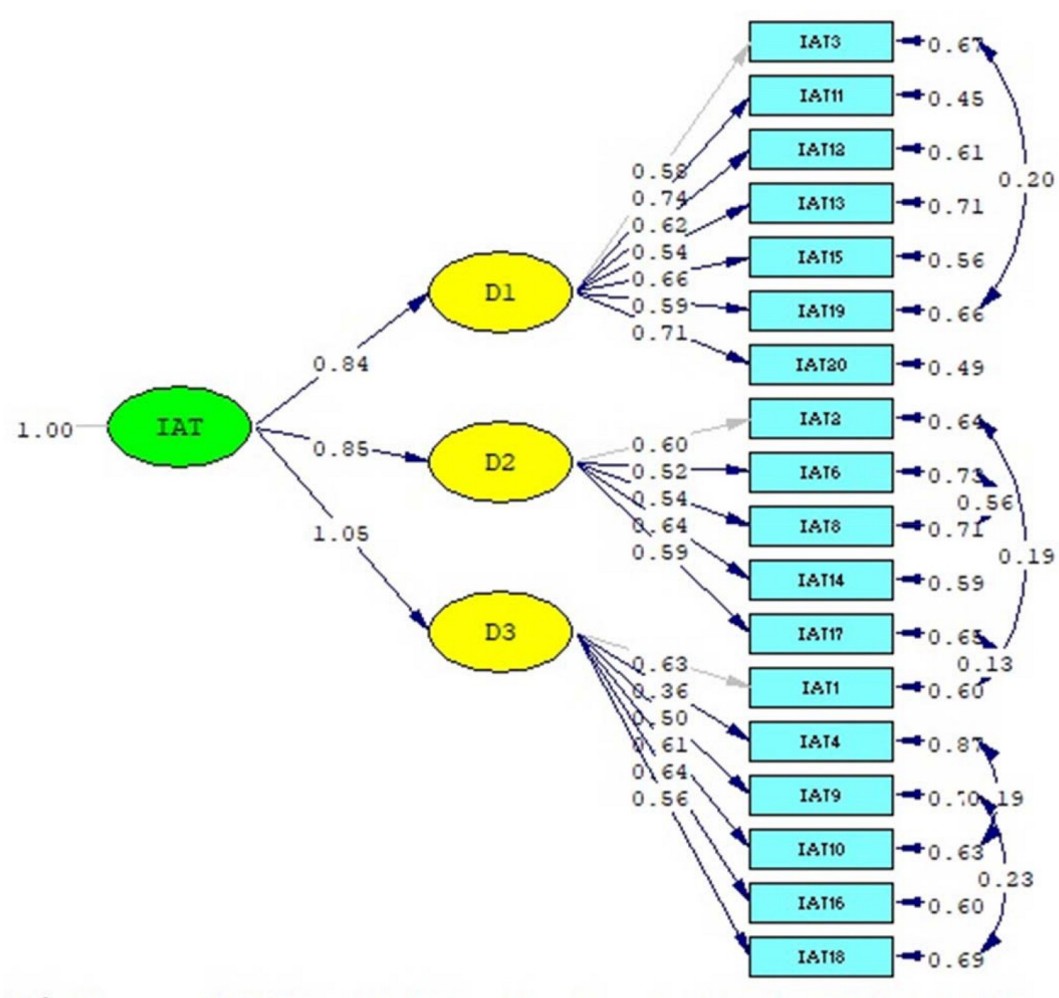

Chi-Square=479.50, df=126, P-value=0.00000, RMSEA=0.066

**Fig 2. Confirmatory factor analysis result of second model IAT (EFA result).**

## Discussions

The primary objective of this study was to examine the psychometric properties of the Indonesian version of IAT. As a part of the validation process, translation and cultural adaptation were undertaken at the beginning to ensure that all items in the Indonesian version of the IAT questionnaire could be understood and effectively perceived by the Indonesian community. All inputs from experts and respondents were thoroughly considered, resulting in the final version of the Indonesian IAT that was further exposed to validity and reliability testing.

**Table 4. Comparison of goodness of fit indices in two models.**

| Model | $x^2$ | df | $x^2$/df | RMSEA | CFI | AIC | SRMR | TLI | AGFI |
|---|---|---|---|---|---|---|---|---|---|
| Model 1 (20 items) | 488.05 | 152 | 3.21 | 0.059 | 0.97 | 604.05 | 0.046 | 0.97 | 0.90 |
| Model 2 (18 items, 3 domain) | 479.50 | 126 | 3.81 | 0.066 | 0.96 | 596.50 | 0.048 | 0.96 | 0.90 |

$x^2$ = Chi-Square, df = Degree of Freedom, RMSEA = Root Mean Square Error of Approximation, CFI = Comparative Fit Index, AIC = Akaike Information Criterion, SRMR = Standardized Root Mean Square Residual, TLI = Non-Normed Fit Index (NNFI), and AGFI = Adjusted Goodness of Fit Index.

**Table 5. Cut-off scores for 18-item Indonesian version of IAT.**

| Level of internet addiction | 20-item original version of IAT [7] | 18-item Indonesian version of IAT |
|---|---|---|
| Normal | 0–30 | 0–27 |
| Mild addiction | 31–49 | 28–44 |
| Moderate addiction | 50–79 | 45–71 |
| Severe addiction | 80–100 | 72–90 |

Upper score of 18-item Indonesian IAT = upper score of 20-item original IAT/20 x 18.

Lower score of 18-item Indonesian IAT = lower score of 20-item original IAT/20 x 18.

The first validity and reliability tests were performed in the pilot study. The test was conducted with 385 subjects. Item number 7, which pertained to how often the subject checks their email before doing other activities, was subsequently excluded due to its poor validity ($r < 0.3$); this was similar to the findings of a prior study in Spanish and can perhaps be explained by the fact that the subjects of the study are junior and senior high school students who rarely use their email accounts for daily activities such as for academic purposes. Moreover, the behavior of email-checking can be regarded as normal in the current era due to easier access to email by smartphones—a feature that was not available at the time of the original IAT questionnaire creation [26].

As per the EFA results, item number 5—pertaining to whether people in the subject's life complain about the amount of time that the subject spends online—has a factor loading $< 0.4$. This result is in accordance with a study conducted in China that claimed that item number 5 has a low diagnostic accuracy value compared to the other items [34]. This could be due to some subjects continuing to browse the internet without the knowledge of the people surrounding them to avoid complaints or prohibition. The majority of respondents in this study live with their parents and utilise the internet connection available in the house. Interviews with several subjects' parents revealed that they did not prohibit their children from using the internet in the house as they felt safer when their children stayed in the house rather than when they played outside. Previous studies also revealed that deceptive behavior by adolescents depends on their parents' attitude towards their playing behavior. A condemning attitude from parents tends to make children lie about their internet use [35].

Our EFA results suggested that the three-factor model of IAT with 18 items has a total variance of 46.392%. Hence, in CFA, we compared this three-factor model with the original version of IAT with a single factor of 20 original items of IAT. Our analysis revealed that three-dimensional IAT displays better psychometric properties than the one-factor model. Prior studies indicated that several IAT models are one-factor to six-factor models [19,23–31]. The variability of the models could be due to diversity in the subjects' characteristics and cultural backgrounds [23,25,27]. However, the same results were also found in other populations; the three-factor solution model was found to be most suitable among Thai, British, Greek, and Iranian samples [23,31,36,37]. For our three-dimensional Indonesian IAT model, internal reliability was evaluated using Cronbach's alpha. The internal consistency score was 0.855 and thereby indicates high reliability of the questionnaire.

Items within the salience domain (item 3, 11, 12, 13, 15, 19, and 20) in this study mostly covered items included in the withdrawal symptoms domain (item 11,12, 13,15, 19, and 20) in the Thailand study [31]. The diversity in this domain can be attributed to the variations in respondent characteristics in the study. However, previous studies indicate a relationship between salience and withdrawal. Internet addicts have salience symptoms with a pre-occupation of using the internet. Thus, if they stop browsing the internet, the withdrawal symptoms

will occur within hours to days [35]. On the other hand, other studies have shown that the items belonged to different domains such as the psychological and emotional conflict domain (item 3, 11, and 19) and time management issue domain (item 12, 13, and 15) in Britain and the psychological and emotional conflict domain (item 3, 15, 19, and 20) and neglected work domain (item 11, 12, and 13) in Greece. In Iran, the items in the salience domain were included in the emotional and mood disorder domain (item 11,12, and 13) and the personal activities disorder domain (item 12, 19, and 20) [23,36]. The second domain in this study is neglect of duty. This result is in accordance with the findings of a previous study in which the items contained in the neglect of duty domain (item 2, 6, 8, 14, and 17) in the study also included underperformance problems (item 2, 6, 8, and 14 in the Thailand study), neglect of work (item 6 and 8 in the Greece Study), and personal activity impairments (item 6, 8, and 14 in the Iran study) [31,36,37]. Interestingly, this domain can also be combined with time management domain in Greece (item 2, 4, and 17) and in Hongkong (item 14), and social problems in Iran (item 2) and in Hongkong (item 2, 6, 8, and 17) [26–28,30,31]. The third domain in the study is loss of control (item 1, 4, 9, 10, 16, and 18). These items included belongs to performance problems and relationship problems (item 1, 4, 9, and 10) in Thailand, and withdrawal symptoms domains in the Thailang (item 18) and Hongkong (item 1, 10, and 16) studies [30,31]. The link between the domains can be explained by the fact that individuals with internet addiction can also exhibit tolerance and withdrawal symptoms that result in uncontrolled internet use behavior and can eventually lead to the neglect of their work and damage interpersonal relationships [35,38,39].

Additionally, in this study, we also found a significant association between the level of internet addiction and both genders and duration of daily internet use. The association found between the extent of internet addiction and gender was in accordance with the conclusion of previous studies. More men were reported in the internet addiction group [23,40,41]. However, in other studies, opposing results were suggested [24,25,42,43]. The differences in the findings across the studies might be due to the variability of the subjects' characteristics in each study [23,25,27].

On the other hand, a significant association between IAT scores and the duration of daily internet use was revealed in this study; this complements previous research findings [23,25,30,40]. We found that 27.5% of the subjects with internet addiction used the internet for more than two hours per day, which exceeds the recommendations of the American Academy of Pediatrics (AAP) that defined two hours as the cut-off for excessive daily media use among children and adolescents [44]. Whether the longer duration of internet use causes the subject's internet addiction or conversely, whether the subject's internet addiction results in longer durations of internet use is still questionable. Thus, further studies are required to determine the causal point of this relationship.

The mean age of the subjects in this study is 14.5 (SD ± 1.67) years. No significant relationship between IAT scores and age was observed, and this is in accordance with prior studies [23,26,40]. The average age at which subjects in the study used the internet for the first time was 10 (SD ± 2.35) years. This was similar to the findings of studies in European countries that reported the age of first internet use at 8 years old [45]. Such early age of internet use has been associated with severe internet addiction [46]. Incongruent with previous findings, the total IAT score in our research did not indicate any significant association with the age of first internet use.

There were some limitations in this study. First, the subjects in this study all grew into adolescence in one city—Jakarta. As the capital city of Indonesia, populations in Jakarta can be considered as the best representation of all of the Indonesian population due to its diversity. In the long run, it would be better if future research is conducted in other regions of Indonesia as

well. Second, there was partial overlapping of participants between EFA and CFA. Third, other forms of reliability testing (e.g. test-retest reliability, parallel form) are also important to be evaluated in future studies. Fourth, due to the nature of self-reporting questionnaires, the existence of recall bias and social desirability were unavoidable. Last, the explained variance was less than 60% in the final Indonesian version of IAT, the following users need to interpret the score of IAT very carefully. Nevertheless, as per our preliminary investigations, this study is a novel study that investigated the psychometric properties of the Indonesian version of IAT.

## Conclusion

In conclusion, the Indonesian version of the IAT demonstrated good validity and reliability in the three-dimensional model. The IAT can be used as a tool for screening internet addiction in the Indonesian population. A significant association between the level of internet addiction and gender and daily internet use duration was also revealed in the study.

## Supporting information

**S1 File.**
(PDF)

## Acknowledgments

The authors would like to thank all of the participants in this study.

## Declarations

**Ethics approval and consent to participate**

Research Ethics Committee of the Faculty of Medicine of Universitas Indonesia - Cipto Mangkusumo Hospital, Jakarta, Indonesia (Reference Number: 318/UN2.F1/ETIK/2016).

Written informed consent was obtained from each of the study participants and their parents or legal guardians.

## Author Contributions

**Conceptualization:** Kristiana Siste, Martina Wiwie Nasrun, Saptawati Bardosono, Rini Sekartini, Riza Sarasvita, Tjhin Wiguna.

**Data curation:** Kristiana Siste, Saptawati Bardosono, Belinda Julivia Murtani, Tjhin Wiguna.

**Formal analysis:** Kristiana Siste, Christiany Suwartono, Reza Damayanti, Tjhin Wiguna.

**Investigation:** Kristiana Siste, Tjhin Wiguna.

**Methodology:** Kristiana Siste, Martina Wiwie Nasrun, Tjhin Wiguna.

**Resources:** Tjhin Wiguna.

**Software:** Christiany Suwartono.

**Supervision:** Saptawati Bardosono, Rini Sekartini, Jacub Pandelaki, Riza Sarasvita, Tjhin Wiguna.

**Validation:** Christiany Suwartono, Belinda Julivia Murtani, Tjhin Wiguna.

**Writing – original draft:** Kristiana Siste, Belinda Julivia Murtani, Tjhin Wiguna.

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
