## [Decision Letter · Decision Letter 0]

4 Sep 2020

PONE-D-20-06905

Validation Study of the Indonesian Internet Addiction Test among Adolescents

PLOS ONE

Dear Dr. Wiguna,

Thank you for submitting your manuscript to PLOS ONE. After careful consideration, we feel that it has merit but does not fully meet PLOS ONE’s publication criteria as it currently stands. Therefore, we invite you to submit a revised version of the manuscript that addresses the points raised during the review process.

An expert in the field of psychometric testing has provided detail comments for you to improve your work. Please kindly review all the comments and addressed them appropriately in the revision. Apart from the comments provided by the reviewers, please be careful for the term of "internet use". Currently, there is no consensus in the the term of using "internet addiction". Therefore, a variety of terms (e.g., internet addiction disorder, problematic internet use, internet addiction.) have been used and indicated to the same thing. Therefore, please also clearly define your term in the revised manuscript. Please see the following papers for your use.

Leung, H., Pakpour, A. H., Strong, C., Lin, Y.-C., Tsai, M.-C., Griffiths, M. D., Lin, C.-Y., Chen, I.-H. (2020). Measurement invariance across young adults from Hong Kong and Taiwan among three internet-related addiction scales: Bergen Social Media Addiction Scale (BSMAS), Smartphone Application-Based Addiction Scale (SABAS), and Internet Gaming Disorder Scale-Short Form (IGDS-SF9)(Study Part A). Addictive Behaviors, 101, 105969.

Chen, I.-H., Strong, C., Lin, Y.-C., Tsai, M.-C., Leung, H., Lin, C.-Y., Pakpour, A. H., Griffiths, M. D. (2020). Time invariance of three ultra-brief internet-related instruments: Smartphone Application-Based Addiction Scale (SABAS), Bergen Social Media Addiction Scale (BSMAS), and the nine-item Internet Gaming Disorder Scale- Short Form (IGDS-SF9) (Study Part B). Addictive Behaviors, 101, 105960. 

Montag, C., Wegmann, E., Sariyska, R., Demetrovics, Z., & Brand, M. (2019). How to overcome taxonomical problems in the study of Internet use disorders and what to do with "smartphone addiction"?. Journal of behavioral addictions, 1–7. Advance online publication. https://doi.org/10.1556/2006.8.2019.59

We look forward to receiving your revised manuscript.

Kind regards,

Chung-Ying Lin

Academic Editor

PLOS ONE

Journal Requirements:

2.In your Data Availability statement, you have not specified where the minimal data set underlying the results described in your manuscript can be found. PLOS defines a study's minimal data set as the underlying data used to reach the conclusions drawn in the manuscript and any additional data required to replicate the reported study findings in their entirety. All PLOS journals require that the minimal data set be made fully available. For more information about our data policy, please see http://journals.plos.org/plosone/s/data-availability.

Reviewers' comments:

Reviewer's Responses to Questions

**Comments to the Author**

1. Is the manuscript technically sound, and do the data support the conclusions?

Reviewer #1: Partly

2. Has the statistical analysis been performed appropriately and rigorously? 

Reviewer #1: No

3. Have the authors made all data underlying the findings in their manuscript fully available?

Reviewer #1: Yes

4. Is the manuscript presented in an intelligible fashion and written in standard English?

Reviewer #1: Yes

5. Review Comments to the Author

Reviewer #1: Major issues:

1. Please explain whether the participants in different stage of process were overlapping.

2. Please add the description about the factor structure of the IAT in the instrument section.

3. The details of the revising process of the IAT need to be provided. For example, how many rounds of the “face validity” did the authors conduct to collect suggestions, revise the test and re-test.

4. The factors in a scale are often related. Please use "oblique rotation" and re-conduct the EFA.

5. It is not recommended to delete the item due to a single reason. Please put all items into the EFA (oblique rotation) and determine whether the items need to be removed.

6. The explained variance of the EFA is < 60%, indicating that the remained items are not sufficient.

7. Page 19 Line 295 “for each of the 19 questionnaire items” Please check how many items were used in the process.

8. The comparison of factor structure of the IAT in different language versions is essential. Please provide more information about the comparison of factor structure of the IAT in different language versions, especially the item allocation.

Miner issues:

1. There are some typos in the manuscript, such as “the refore”. Please correct them.

2. Page 14 Line175 “The English version of the instrument was subsequently assessed by three experts, including an addiction psychiatrist,…” is confusing, because the authors had translated the IAT to into Indonesian version at the previous stage. Please rephrase the sentence.

3. It is less common to apply a large number of participants in a pilot study. Is there any concern of author to do so?

4. Please change the name “Statistical Package for the Social Sciences (SPSS) version 22”to "IBM Statistical Package for the Social Sciences (SPSS) version 22."

5. The “face validity” in this study is more like “item analysis”. Please change the term.

6. Using the internet is not a disease. Therefore, it is inappropriate to say “onset of internet use”.

6. PLOS authors have the option to publish the peer review history of their article (what does this mean?). If published, this will include your full peer review and any attached files.

Reviewer #1: No

---

## [Author Response · Author response to Decision Letter 0]

12 Nov 2020

Comments Responses

1- Please explain whether the participants in different stage of process were overlapping. 

Thank you for the comment. We have tried to ensure that participants do not overlap in each stage, but there are indeed overlapping participants at the EFA and CFA stages. Although the participants in the CFA did not completely overlap with the EFA, there are new participants included in the CFA. In addition, there is an advantage that these two stages have a large number of participants.

2- Please add the description about the factor structure of the IAT in the instrument section. 

Thank you for the input. We have added the description about the factor structure of the IAT in line 167 – 169.

3- The details of the revising process of the IAT need to be provided. For example, how many rounds of the “face validity” did the authors conduct to collect suggestions, revise the test and re-test. 

Thank you for the comment. The details of the revising process of the IAT is already written in the manuscript. However, to make it clearer, we have revised it to “one round of item analysis was conducted on 15 JHS students and 15 SHS from seven selected schools through the focus group discussion method to determine the comprehensibility and efficiency of the instructions and terms used in the questionnaire.” (line 186).

4- The factors in a scale are often related. Please use "oblique rotation" and re-conduct the EFA. 

Thank you for the comment. We have re-conducted the EFA using oblique rotation (direct oblimin and promax), the result is:

a) direct oblimin

The final result showed IAT consists of only two factors and ten items. The total variance is 45.99% and factor loads ranged from 0.464 – 0.842.

b) promax

The final result showed IAT consists of three factors and fifteen items. The total variance is 49.34% and factor loads ranged from 0.422 – 0.893.

It indicates that EFA using oblique rotation is not any better than EFA we have conducted in the manuscript (using varimax). Therefore, may we be please allowed to keep the EFA result as we have contucted before in our manuscript?

5- It is not recommended to delete the item due to a single reason. Please put all items into the EFA (oblique rotation) and determine whether the items need to be removed. 

Thank you for the comment. In accordance with your advice, we have put all items into the EFA using oblique rotation (direct oblimin and promax) and put all items, but we get the same result as before. In both of oblique rotation types, the two items (item 5 and 7) we deleted in the manuscript are still need to be removed (the result showed those items have loading factors <0.4). In addition, the two items we deleted in the manuscript due to the qualitative and quantitative reasons. The quantitative reason is because of the loading factor <0.4, whereas the qualitative reason is already explained in the discussion section (line 348-366). Therefore, we decided to delete those items.

6- The explained variance of the EFA is < 60%, indicating that the remained items are not sufficient. 

Thank you for the comment. The variance describes how well the items in the IAT measure internet addiction, and the variance will decrease when the item is removed. A variance of less than 60% indicates it is likely that more factors emerged than the expected factors in the model. Therefore, it needs to be accounted for that internet addiction assessment is not only assessed by the IAT. Internet addiction is also influenced by many factors, for example, certain demographic characteristics. In addition, the IAT is not a questionnaire developed in Indonesia so it is possible that the items in the IAT are not sufficient to describe internet addiction in Indonesia when used by Indonesian. We are aware of the limitation of the IAT psychometric in this study, so it is needed to build a questionnaire based on the demographic characteristics in Indonesia.

7- Page 19 Line 295 “for each of the 19 questionnaire items” Please check how many items were used in the process. 

Thank you for the comment. We have revised it to “for each of the 18 questionnaire items” (line 298). The Indonesian version of IAT consists of 18 items after going through the reliability and the EFA process.

8- The comparison of factor structure of the IAT in different language versions is essential. Please provide more information about the comparison of factor structure of the IAT in different language versions, especially the item allocation. 

Thank you for the comment. We already made the comparison between the Indonesian version of IAT with other different language versions of IAT in the discussion section. However, following your advice, we had added more information about the item allocation (line 378 – 400). We also had compared every item in each domain of Indonesian version of IAT with IAT in different language versions.

---

## [Decision Letter · Decision Letter 1]

8 Dec 2020

PONE-D-20-06905R1

Validation Study of the Indonesian Internet Addiction Test among Adolescents

PLOS ONE

Dear Dr. Wiguna,

Thank you for submitting your manuscript to PLOS ONE. After careful consideration, we feel that it has merit but does not fully meet PLOS ONE’s publication criteria as it currently stands. Therefore, we invite you to submit a revised version of the manuscript that addresses the points raised during the review process.

We look forward to receiving your revised manuscript.

Kind regards,

Chung-Ying Lin

Academic Editor

PLOS ONE

Reviewers' comments:

Reviewer's Responses to Questions

**Comments to the Author**

1. If the authors have adequately addressed your comments raised in a previous round of review and you feel that this manuscript is now acceptable for publication, you may indicate that here to bypass the “Comments to the Author” section, enter your conflict of interest statement in the “Confidential to Editor” section, and submit your "Accept" recommendation.

Reviewer #1: (No Response)

2. Is the manuscript technically sound, and do the data support the conclusions?

Reviewer #1: Partly

3. Has the statistical analysis been performed appropriately and rigorously? 

Reviewer #1: No

4. Have the authors made all data underlying the findings in their manuscript fully available?

Reviewer #1: No

5. Is the manuscript presented in an intelligible fashion and written in standard English?

Reviewer #1: Yes

6. Review Comments to the Author

Reviewer #1: Most comments have been addressed.

1. I still think the participants in EFA and CFA should not be the same. If you separate the participants into two groups, will the sample size be too small to conduct the EFA and CFA? If the answer is yes, please add the description about how many participants overlapped in EFA and CFA in the results. Moreover, please add a limitation about this technical issue.

2. Please add a clearly caution that the explained variance of the EFA is < 60%, indicating that the remained items in the IAT are not sufficient. The following users need to interpret the score of IAT very carefully. Adding more items into the IAT is recommended.

7. PLOS authors have the option to publish the peer review history of their article (what does this mean?). If published, this will include your full peer review and any attached files.

Reviewer #1: No

---

## [Author Response · Author response to Decision Letter 1]

8 Jan 2021

Comments Responses

1- I still think the participants in EFA and CFA should not be the same. If you separate the participants into two groups, will the sample size be too small to conduct the EFA and CFA? If the answer is yes, please add the description about how many participants overlapped in EFA and CFA in the results. Moreover, please add a limitation about this technical issue. Thank you for the comment. We have added it as a part of our limitations in this study. 

Line 368: “Second, there was partial overlapping of participants between EFA and CFA.”

We also have mentioned the number of participants that overlapped in EFA and CFA in the results section. 

Line 242-243: “To note, the first 385 respondents in the dataset were overlapping and employed within EFA (N= 385) and CFA (N= 643). 

2- Please add a clearly caution that the explained variance of the EFA is < 60%, indicating that the remained items in the IAT are not sufficient. The following users need to interpret the score of IAT very carefully. Adding more items into the IAT is recommended. Thank you for the input. We have stated it as a limitation within the revised manuscript:

Line 371-373: “Last, the explained variance was less than 60% in the final Indonesian version of IAT, the following users need to interpret the score of IAT very carefully.”

---

## [Editor Report · Decision Letter 2]

11 Jan 2021

Validation Study of the Indonesian Internet Addiction Test among Adolescents

PONE-D-20-06905R2

Dear Dr. Wiguna,

We’re pleased to inform you that your manuscript has been judged scientifically suitable for publication and will be formally accepted for publication once it meets all outstanding technical requirements.

Kind regards,

Chung-Ying Lin

Academic Editor

PLOS ONE
---

## [Editor Report · Acceptance letter]

19 Jan 2021

PONE-D-20-06905R2 

Validation Study of the Indonesian Internet Addiction Test among Adolescents 

Dear Dr. Wiguna:

I'm pleased to inform you that your manuscript has been deemed suitable for publication in PLOS ONE. Congratulations! Your manuscript is now with our production department. 

Kind regards, 

on behalf of

Dr. Chung-Ying Lin 

Academic Editor

PLOS ONE